# Primary Pulmonary B-Cell Lymphoma: A Review and Update

**DOI:** 10.3390/cancers13030415

**Published:** 2021-01-22

**Authors:** Francesca Sanguedolce, Magda Zanelli, Maurizio Zizzo, Alessandra Bisagni, Alessandra Soriano, Giorgia Cocco, Andrea Palicelli, Giacomo Santandrea, Cecilia Caprera, Matteo Corsi, Giulia Cerrone, Raffaele Sciaccotta, Giovanni Martino, Linda Ricci, Francesco Sollitto, Domenico Loizzi, Stefano Ascani

**Affiliations:** 1Pathology Unit, Azienda Ospedaliero-Universitaria, Ospedali Riuniti di Foggia, 71122 Foggia, Italy; 2Pathology Unit, Azienda USL-IRCCS di Reggio Emilia, 42122 Reggio Emilia, Italy; Magda.Zanelli@ausl.re.it (M.Z.); Alessandra.Bisagni@ausl.re.it (A.B.); Andrea.Palicelli@ausl.re.it (A.P.); Giacomo.Santandrea@ausl.re.it (G.S.); 3Surgical Oncology Unit, Azienda USL-IRCCS di Reggio Emilia, 42122 Reggio Emilia, Italy; Maurizio.Zizzo@ausl.re.it; 4Clinical and Experimental Medicine PhD Program, University of Modena and Reggio Emilia, 41121 Modena, Italy; 5Gastroenterology, Division and Inflammatory Bowel Disease Center, Department of Internal Medicine, Azienda USL-IRCCS di Reggio Emilia, 42122 Reggio Emilia, Italy; Alessandra.Soriano@ausl.re.it; 6Radiotherapy Unit, Azienda Ospedaliero-Universitaria, Ospedali Riuniti di Foggia, 71122 Foggia, Italy; gcocco@ospedaliriunitifoggia.it; 7Pathology Unit, Azienda Ospedaliera S. Maria di Terni, University of Perugia, 05100 Terni, Italy; ceciliacaprera@libero.it (C.C.); matteocorsi81@gmail.com (M.C.); gi.cerrone@gmail.com (G.C.); sciaccottaraffaele@gmail.com (R.S.); gio.martino@gmail.com (G.M.); lindaricci1987@hotmail.it (L.R.); s.ascani@aospterni.it (S.A.); 8Institute of Thoracic Surgery, University of Foggia, 71122 Foggia, Italy; francesco.sollitto@unifg.it (F.S.); domenico.loizzi@unifg.it (D.L.)

**Keywords:** pulmonary B-cell lymphoma, BALT, MALT lymphoma, diffuse large B-cell lymphoma, lymphomatoid granulomatosis, primary effusion lymphoma, intravascular large B-cell lymphoma

## Abstract

**Simple Summary:**

The group of B-cell lymphomas primarily involving the lung encompasses different histological entities with distinct biological aspects, while sharing some clinical and radiological features related to their common anatomic site of occurrence. Recent molecular advances in the molecular genetics of these lesions have substantially improved of our understanding of the mechanisms of lymphomagenesis, adding novel information to histology in order to better characterize and manage these diseases. This review summarizes the available clinical, radiological, pathological, and molecular data on primary pulmonary B-cell lymphomas, discusses the mechanisms of lymphomagenesis, and highlights the role of a multi-disciplinary management in overcoming the diagnostic and therapeutic challenges in this setting.

**Abstract:**

Primary pulmonary B-cell lymphomas (PP-BCLs) comprise a group of extranodal non-Hodgkin lymphomas of B-cell origin, which primarily affect the lung without evidence of extrapulmonary disease at the time of diagnosis and up to 3 months afterwards. Primary lymphoid proliferations of the lung are most often of B-cell lineage, and include three major entities with different clinical, morphological, and molecular features: primary pulmonary marginal zone lymphoma of mucosa-associated lymphoid tissue (PP-MZL, or MALT lymphoma), primary pulmonary diffuse large B cell lymphoma (PP-DLBCL), and lymphomatoid granulomatosis (LYG). Less common entities include primary effusion B-cell lymphoma (PEL) and intravascular large B cell lymphoma (IVLBCL). A proper workup requires a multidisciplinary approach, including radiologists, pneumologists, thoracic surgeons, pathologists, hemato-oncologists, and radiation oncologists, in order to achieve a correct diagnosis and risk assessment. Aim of this review is to analyze and outline the clinical and pathological features of the most frequent PP-BCLs, and to critically analyze the major issues in their diagnosis and management.

## 1. Introduction

Primary pulmonary B-cell lymphoma (PP-BCL) is a rare entity, representing less than 1% of all non-Hodgkin lymphoma (NHL), and 3–4% of all extranodal NHL [1,2,3]. Overall, PP-BCL is more frequent in adults (median age, 60 years), and is rare in younger patients [4].

Spread of systemic lymphoma to the lung by hematogenous dissemination or contiguous invasion through extension from hilar or mediastinal nodes is much more common, occurring in up to 24% of patients with NHL [5], and usually associated with an unfavorable prognosis [6,7,8,9]. In order to differentiate a PP-BCL from secondary pulmonary involvement by systemic lymphoma, the lack of extrapulmonary disease at presentation, and over the course of the subsequent three months should be confirmed by comprehensive clinical, radiologic, and pathologic assessment [4,9].

Primary lymphoproliferative processes affecting the parenchyma and/or bronchi of one or both lungs arise from the mucosa-associated lymphoid tissue specific to the lung, or bronchus-associated lymphoid tissue (BALT), which is one component of the complex pulmonary lymphatic system [10]. The most frequent entity of PP-BCL is extranodal marginal zone lymphoma of mucosa-associated lymphoid tissue (PP-MZL, or MALT lymphoma), with other entities, including other subtypes of low-grade lymphoma, lymphomatoid granulomatosis (LYG) and primary pulmonary diffuse large B cell lymphoma (PP-DLBCL), being far less common [8,9]. PP-BCL involving the pleural cavity, such as primary effusion lymphoma (PEL) is rare, as well as intravascular large B cell lymphoma (IVLBCL) [11] (Table 1).

In this article, we will discuss the pathogenesis of the most typical B-cell lymphoproliferative disorders primarily involving the lung, namely, PP-MZL/MALT lymphoma, PP-DLBCL, PEL, IVLBCL, and LYG; furthermore, we will review their epidemiological features, clinical presentation, pathological and molecular aspects, along with prognostic factors and therapeutic options; finally, we will critically analyze the major issues in their diagnosis and management.

## 2. Primary Pulmonary Marginal Zone Lymphoma of Mucosa-Associated Lymphoid Tissue (PP-MZL/MALT Lymphoma)

### 2.1. Bronchus-Associated Lymphoid Tissue (BALT) and Lymphomagenesis

The lung parenchyma features many lymphatics and a lymphoid compartment, the latter being sorted into distinct categories: (1) the system of hilar and intra-parenchymal lymph nodes, (2) the mucosa-associated lymphoid tissue specific to the lung, or bronchus-associated lymphoid tissue (BALT), and (3) peripheral lymphoid aggregates, single lymphocytes and phagocytes close to bronchi and bronchioli [5,7].

Resting BALT is a system of nodular aggregates of lymphoid tissue mainly composed of poorly formed primary follicles, mantle and marginal zones, and interfollicular regions [12]; it is located throughout the airways, mainly at distal bronchial and bronchiolar bifurcations, where it functions as a barricade against airborne antigens and infectious agents [13]. BALT is composed primarily of B cells at the center with peripheral clusters of T cells [14] and drains through efferent lymphatic channels to regional lymph nodes [15]. The lymphoid elements in BALT are arranged within a reticular stroma associated to a specialized pseudostratified bronchial epithelium depleted in cilia and goblet cells and infiltrated by T lymphocytes; such ‘lymphoepithelium’ is capable of specific phago-pinocytosis and antigen presentation [11,15].

Upon exposure to inhaled particles and/or intrinsic or extrinsic antigenic stimulation, a sequence of immunological processes ensues, involving lymphoepithelial cells, antigen-presenting cells, and naive B and T cells [7,15]. Such ‘BALT induction’ leads to several morphological and functional changes, namely the formation of true germinal centers within follicles, composed of actively proliferating and differentiating B cells (secondary follicles), the development of visible mantle zones, and the enlargement of the interfollicular region (Figure 1). Functionally, the resulting ‘activated BALT’ is able to release cytokines and produce IgA, and to generate memory T cells and plasma cells, in order to promote an effective immune response [15,16].

In adulthood, chronic antigenic stimulation, or neoplastic transformation of BALT can lead to the development of reactive disorders, namely non-neoplastic lymphoproliferative disorders of the lung, or to malignant lymphomas [8,15,17]. The development of MALT lymphoma ensues through two main pathogenetic steps: (1) an antigen-dependent phase, where clonal expansion of B cells and their survival is linked to chronic exposure to specific antigenic stimuli [18], and (2) an antigen-independent phase, where proliferation of B lymphocytes becomes autonomous as a result of cytogenetic and molecular alterations, which are detectable in approximately 50% of PP-MZLs (see below). Further mutations developing during lymphomagenesis allow the proliferating cells to acquire all the features of a neoplastic cell, thus accomplishing the process of lymphomagenesis [19].

### 2.2. Risk Factors

BALT induction and prolonged exposure to antigens, such as microorganisms and/or autoimmune disorders, represent early steps in the pathogenesis of PP-MZL. MALT lymphoma may arise in several anatomic sites, and sometimes it is possible to identify a specific causative agent on the basis of the development of the disease (for instance, Helicobacter pylori infection in gastric MALT lymphoma) [20]. Conversely, no infectious agent seems to play an etiological role in the pathogenesis of PP-MZL, although infectious and inflammatory processes are frequently present in patients affected by pulmonary MALT lymphoma. It has been hypothesized that such conditions represent the chronic inflammatory stimulation resulting in BALT induction and maintenance and, afterwards, in molecular and morphological changes leading to neoplastic transformation of proliferating B cells [21].

Adam et al. reported on patients affected by PP-MZL from six European countries, and detected *Achromobacter xylosoxidans*, a Gram-negative aerobic motile bacterium, which is an opportunistic pathogen with low virulence and antibiotic-resistance, typically present in patients with cystic fibrosis [22]. Although these results suggest a strong association between *A. xylosoxidans* infection and MALT lymphomagenesis, findings from a Japanese study are contradictory [23], describing a low prevalence of this bacterium in lung biopsies from patients affected by PP-MZL assessed by PCR-based analysis. Conversely, in this study *A. xylosoxidans* was more frequent in biopsies from patients with DLBCL. Such conflicting findings may be partly due to the heterogeneous geographic distribution of this bacterium, resulting in a higher incidence of cystic fibrosis in Western countries [23]. Other studies describe a correlation between *Mycobacterium tuberculosis* infection and the development of PP-MZL, claiming that patients with active tuberculosis not adequately managed by antituberculosis drugs may have an increased risk of developing lymphoma over time, since activated macrophages and T-cells are capable to stimulate clonal proliferation of B lymphocytes [24].

Furthermore, molecular analysis of microbial DNA and/or RNA from biopsy samples of PP-MZL detected in some cases traces of microorganisms, such as *Chlamydophila pneumoniae*, *Chlamydia trachomatis*, *Chlamydia psittaci* and *Mycoplasma pneumoniae* [25].

On the other hand, immune disorders, both autoimmune and immunosuppressive, represent in most cases an optimal anlage for the development and maintenance of PP-MZL, including systemic lupus erythematosus (SLE), multiple sclerosis, and Hashimoto thyroiditis [25]. For instance, a lymphoma occurs in 5–10% of patients affected by Sjögren syndrome (SS), with an estimated risk 4 to 16 times higher than the general population [26]. Moreover, the cumulated risk to develop a lymphoid neoplasm in patients with SS is higher than those with other autoimmune disorders, such as SLE and rheumatoid arthritis (RA). Specific predictive factors have been identified, which affect the risk of lymphomagenesis in patients with SS, including clinical (namely, persistent enlargement of salivary glands, splenomegaly, lymphadenopathy, and palpable purpura) and biological features (cryoglobulinemia, lymphopenia (mainly decrease of CD4+ T cells), urine and serum monoclonal component, presence of rheumatoid factor [26]. The role of rheumatoid factor in the MALT lymphomagenesis of patients affected by SS has been confirmed by other studies [27].

### 2.3. Epidemiology and Clinical Features

MALT lymphoma is the most common type of primary pulmonary lymphoma, accounting for 70% to 90% of all cases [28,29,30]. Lung is the 4th most frequent site of occurrence of extranodal MALT lymphomas in the US after stomach, spleen, and eye, according to a recent large US Surveillance, Epidemiology, and End Results (SEER) registry series, with a rate as high as 7.7% [31], which is similar to the 9% rate reported by the WHO Classification of tumors of hematolymphoid tissues [3]. The same authors reported on a 2-step increase in 18% of the incidence of PP-MZL between 2001 to 2005 and 2006 to 2009 [31].

MALT lymphoma usually affects female patients in their 60–70 s (median age, 67–68 years) [31,32,33], although the slight female predominance is not a consistent finding throughout the literature [31]. The relatively high age of incidence suggests that a long exposure to specific risk factors is indispensable to promote carcinogenesis, however younger individuals with an underlying immunosuppression status may be interested as well [31]. Cigarette smoking may promote the development of a chronic inflammatory anlage within respiratory airways, thus contributing to lymphomagenesis [30], with previous reports on the coexistence of lung cancer and pulmonary MALT lymphoma in heavy smokers [20]. Autoimmune diseases, such as SS, RA, SLE, may be present in approximately one third of cases [7,29,34] (see above). Data from a series of 41 PP-MZLs and 9 pulmonary mixed lymphomas (MALT lymphoma + DLBCL) support the higher incidence of autoimmune disorders in the group of pure MALT lymphoma compared to the group with mixed histology [29]; however, the interpretation of such results may be questioned as the so-called ‘mixed lymphomas’ today would be probably classified as MALT lymphomas progressed to DLBCL, rather than distinct histotypes coexisting within the same organ.

Approximately 30% of patients are asymptomatic, with the remaining presenting with systemic or organ-related symptoms, such as fever, malaise cough, dyspnea, and less frequently hemoptysis and chest pain [11,30,35]; the coexistence of fever and weight loss should raise the suspicion of an aggressive disease form [9,36]. Due to non-specificity of its clinical presentation, PP-MZL is often misdiagnosed as a pneumonic process persisting despite antibiotics [4].

Serum findings include monoclonal gammopathy detected by serum protein electrophoresis (<3 g/dL) in up to 60% of patients [29], and more often in cases with plasmacytic differentiation and disseminated disease [30], with a serum monoclonal IgM spike in 25–30% of cases, and less commonly, involvement of IgG and IgA classes [37]. The same monoclonal light chain types are detected in both serum and neoplastic cells. Other serum findings, such as LDH levels, are not contributory.

### 2.4. Imaging

Chest computed tomography (CT) is the imaging modality of choice for PP-MZL. At imaging, pulmonary MALT lymphomas often present as incidental solitary or multiple nodules of less than 5 cm in size in one lung, mimicking other entities, such as organizing pneumonia, eosinophilic pneumonia, alveolar pattern of sarcoidosis, multifocal adenocarcinoma/adenocarcinoma in situ, nodular lymphoid hyperplasia (NLH), and lymphoid/lymphocytic interstitial pneumonia (LIP) [10]. The wide variability and low specificity of imaging findings often results in a delayed diagnosis, with the time between the initial abnormal clinical/radiological findings and diagnosis ranging from 15 days to 8 years (mean, 9 months) [30,37].

Both lungs are involved in approximately one quarter of cases [4,37], thus simulating LIP [38]. Other possible findings are, (1) ill-defined infiltrates with CT attenuation (mixed or pure ground-glass appearance) containing air bronchograms [39], (2) a peripheral mass with pleural thickening [40,41], (3) consolidated lung parenchyma (up to an entire lobe), with air bronchograms and areas of apparent cavitation (bubble-like radiolucencies) [5,11,35,39], suggesting a pneumonic or infectious process [25,42], (4) endobronchial presentation, which may be localized, multifocal, or diffuse throughout the bronchial and tracheal walls [43], and finally (5) along the small airways, with a mosaic attenuation pattern which may mimic other non-neoplastic diseases such as hypersensitivity pneumonitis [44]. Perihilar thickening through bronchovascular bundles across an interlobar fissure, known as the ‘butterfly sign’, has been suggested as a potential radiologic clue to MALT lymphoma over other types of lung lymphomas, along with the ‘bronchogram and CT-angiogram signs’ (an enhancing vessel on a background of consolidation) [42]. In most patients, lesions tend to be stable in size over several years at follow-up CT imaging [38]. Hilar and/or mediastinal lymphadenopathy is infrequently detected (about 30% of cases) [5]. Bronchiectasis can be often seen in larger lesions; conversely, necrosis is not common [38], as well as reticulonodular opacities, atelectasis, and pleural effusions; the latter are seen in 10% of cases [5,37,39].

Fluorodeoxyglucose positron emission tomography (FDG-PET) shows high sensitivity in this setting, with avidity rates as high as 80–100% [29,37,44], usually with a significant association with tumor size [45]; however, its use for staging purposes remains controversial [39].

### 2.5. Pathology

Grossly, PP-MZL presents as a tan-white to light grey homogeneous, not encapsulated mass with relatively well-defined borders and peripheral areas of consolidation. The cut surface is firm, fibrous or granular [20]. The presence of cystic areas is a typical finding and should raise the suspicion of associated amyloidosis [28]. Airway lumina are usually spared, and the visceral pleura may be involved by peripheral lesions forming either a tan-white thickened plaque or multiple polypoid nodules [4].

At light microscopy, pulmonary MALT lymphoma shows lymphoid infiltrates disrupting the lung architecture (Figure 2A). Such lesions display a spectrum of architectural patterns, ranging from discrete nodular lesions with variably defined borders, to a lymphangitic-type distribution along the pleura, interlobular septa, and around bronchovascular bundles, or a combination of both [4,46]. The classic histological ‘triad’ of MALT lymphoma includes reactive lymphoid follicles, diffuse infiltration by centrocyte-like lymphocytes, and lymphoepithelial lesions [4]. No single feature may be a criterion by itself [4].

The neoplastic cells are heterogeneous, expand the interfollicular regions, and are thought to derive from a single precursor cell with multilineage potential [28] (Figure 2B). They include (1) small centrocyte-like lymphocytes with round, regular to indented ‘cleaved’ nuclei, moderately dispersed chromatin, and scant cytoplasm, (2) monocytoid lymphocytes with moderately condensed chromatin, inconspicuous nucleoli, and more abundant amphophylic-to-pale eosinophilic cytoplasm forming a perinuclear halo [10,29], and variable amount of (3) plasmacytoid lymphocytes and plasma cells. The latter may contain immunoglobulin inclusions, in the form of intracytoplasmic Russell bodies and intranuclear PAS-positive Dutcher bodies [10]. Plasma cells are commonly monotypic, however, admixed reactive polytypic plasma cells are easily identified; plasmacytic differentiation may occasionally be extensive. Scattered large immunoblasts and centroblasts are a common finding, conversely sheets of large lymphoid cells point toward transformation to DLBCL [4,29].

Lymphoepithelial lesions are commonly detectable in MALT lymphoma, as well as in hyperplastic BALT and reactive inflammatory processes, featuring clusters of neoplastic lymphoid cells (≥3) infiltrating and sometimes disrupting the ciliated bronchial and/or bronchiolar epithelium, or sometimes mucosal glands [47]. Broad sheets of centrocyte-like lymphocytes effacing the pulmonary parenchyma favor lymphoma over a reactive condition [4,11]. Erosion of the bronchial cartilage and vascular involvement are frequent findings [20].

Reactive, variable sized lymphoid follicles are usually present within MALT lymphoma [4]. Marginal zone cells and/or several, occasionally monotypic plasma cells within abnormal germinal centers devoid of tingible body macrophages result in follicular colonization, which may be a frequent finding. Less specific features of pulmonary MALT lymphoma encompass vascular invasion without necrosis [4], prominent stromal sclerosis and lung parenchyma consolidation due to confluent neoplastic infiltrate [4], ill-formed non-necrotizing epithelioid granulomas [4,46], occasional massive crystal storing histiocytosis, mostly in overt plasmacytic differentiation [48], large lamellar bodies (rings of eosinophilic proteinaceous material) within the spared alveoli in 20% of cases [29,46], and multinucleated giant cells [49].

Pulmonary MALT lymphomas with prominent plasmacytic differentiation show a frequent association with amyloidosis, paraproteinemia, and light and/or heavy chain deposition [50]. This MALT lymphoma subset is most common in elderly women (mean age, 70 years), usually affected by autoimmune disorders, such as SS and RA. Such patients do not show signs of systemic amyloidosis neither serum M-protein [20].

A prominent plasma cell component as well as amyloid deposits allow to distinguish this subtype from conventional PP-MZL, with both sharing the same morphological and immunophenotypical features. Amyloid deposits are mostly composed of protein fibrils, highlighted by specific stains, such as Congo Red, with a typical apple green birefringence in polarized light. Amyloid deposits are seen within the lymphomatous stroma, and also along vessel and/or bronchiolar walls [20]. Pulmonary MALT lymphoma with amyloidosis should be distinguished from primary lung amyloidosis; the two entities may occasionally coexist. Primary lung amyloidosis may present as three distinct entities: nodular pulmonary amyloidosis (NPA), tracheobronchial amyloidosis (TA), and diffuse alveolar septal amyloidosis (DASA). Among them, NPA is the most often associated with MALT lymphoma [51], featuring nodular eosinophilic amyloid deposits and a lymphoplasmacytic inflammatory infiltrate. When NPA and MALT lymphoma coexist within the same lung, amyloid deposits may mask the underlying lymphoid proliferation [51]. However, comprehensive morphological and immunohistochemical examination of these lesions allow the diagnosis of PP-MZL; useful clues are the detection of aggregates of >20 plasma cells with laminar distribution, pleural involvement, the coexistence of reactive lymphoid follicles, and the presence of aberrant CD20/CD43 coexpression (IHC) [51].

### 2.6. Immunohistochemistry and Molecular Findings

Typically, MALT lymphoma shows positivity for CD19, CD20, CD22, and CD79a, and negativity for CD3, CD5, CD10, CD23, BCL6, and cyclin D1 [3,4,9,29] (Figure 2C,D). BCL2 staining is an adjunct in distinguishing neoplastic cells from reactive monocytoid lymphocytes, however its intensity is lower than in follicular lymphoma (FL) [4,50]. Combined immunostaining for a B-cell marker and/or cytokeratins supports the identification of lymphoepithelial lesions [4,52] (Figure 2E). Although T-cell markers highlight small reactive lymphocytes [53], CD43 and B-cell markers coexpression in 30-80% of cases favors a MALT lymphoma over a reactive lesion [5,53]. CD5, or rarely, BCL6 or CD10 may be aberrantly expressed by the neoplastic cells [3,4], with the caveat that a background of CD10+ B cells corresponding to reactive germinal center cells may be detectable at flow cytometry, leading to misdiagnosis of MALT lymphoma as a reactive lesion; however, only about 50% of MALT lymphomas exhibit a distinct immunophenotypic aberrancy of the B cells.

Myeloid cell nuclear differentiation antigen (MNDA) and the immunoglobulin superfamily receptor translocation-associated 1 (IRTA-1) are common markers of extranodal MALT lymphomas, being mostly expressed in and around the lymphoepithelial lesions [54]; the latter is useful in the differential diagnosis with other small B-cell neoplasms [55]. Conversely, MNDA is often negative in FL (<10%), but can be also positive in some types of small B-cell lymphoma [55].

Immunostain for CD21, CD23, or CD35 may highlight the follicular dendritic cell meshwork of colonized and reactive lymphoid follicles [4,52].

The Ki-67 proliferation index is low in the neoplastic cells (< 10%) but is typically higher in residual germinal centers [4,28,52] (Figure 2F).

Kappa and lambda light chain restriction of plasma cells and/or B lymphocytes by flow cytometry, IHC or in situ hybridization (ISH) may be essential to support the diagnosis of lymphoma; identical light chain can be demonstrated in both cell types. However, cases with few plasma cells are usually polytypic [52]. IgM is the most common class involved, and less frequently IgA and IgG [19]. PCR analysis can also be used to demonstrate a clonal heavy chain and light chain gene rearrangement, also in paraffin-embedded tissue samples with positivity rates as high as about 60%; in this setting, a prominent polyclonal background may be a confounding factor [4].

Up to 50% of MALT lymphomas carry a chromosomal translocation resulting in the production of a chimeric protein (BIRC3-MALT1) and in transcriptional deregulation of BCL10, MALT1, and FOXP1, with variable frequencies depending on the site of occurrence [36,56]. Trisomy of chromosome 3 and 18 are often detected in MALT lymphoma, although non-specific. In both pulmonary and gastric MALT lymphoma, the most common cytogenetic abnormality is the t(11;18)(q21;q21) (31–53%), which generates an API2-MALT1 fusion transcript from the API2 (Apoptosis Inhibitor 2) and MALT1 genes, resulting in activation of the NF-kB pathway, which is a potential therapeutic target [57]; trisomy 3 (20%), t(14;18)(q32;q21) (IGH-MALT1) (up to 10%), trisomy 18 and t(1;14)(p22;q32) (BCL10-IGH) (both up to 7%) may occur with decreasing frequency [45,58,59]. Other genetic alterations, such as abnormalities of TNFAIP3 on chromosome 6q23, which may include deletions, mutations, and promoter methylation, have been reported in 15–30% of cases [60]. MYD88 L265P mutations may occur in a minority of cases of MALT lymphoma (6–9%) at different sites, including lung [61]. Such cytogenetic abnormalities play a pivotal role in lymphomagenesis along with antigenic stimulation, as suggested by studies on animal models [62,63,64].

The majority of these translocations can be detected by interphase fluorescence in situ hybridization (FISH) assays with specific break-apart probes or by RT-PCR, from paraffin-embedded tissues or fresh frozen tissue samples [51,65].

Interestingly, gene expression profiling has demonstrated that lung MALT lymphomas have a prominent T-cell signature, show upregulation of marginal zone/memory B cells-associated genes, and plasmacytic differentiation is more common in cases lacking the t(11;18) translocation [66]. Further genetic alterations assessed by next generation sequencing analysis, mostly involving chromatin remodeling, BCR/NF-kB and NOTCH pathways, along with recurrent inactivations of TET2 have been recently described [67].

### 2.7. Differential Diagnosis

A number of benign reactive lymphoid processes and B-cell lymphomas enter the differential diagnosis of a PP-MZL, including diffuse lymphoid hyperplasia (DLH)/LIP, NLH, follicular bronchiolitis, chronic non-specific inflammatory reaction, and chronic lymphocytic leukemia/small lymphocytic lymphoma (CLL/SLL), mantle cell lymphoma (MCL), low-grade follicular lymphoma (FL), lymphoplasmacytic lymphoma (LPL), respectively [52,68]. A thorough clinical history strongly supports the distinction between PP-MZL and a secondary involvement from the stomach, salivary gland, or lymph nodes [52].

The differential diagnosis between reactive lymphoid lesions and MALT lymphoma may be challenging, due to their overlapping morphology (namely, a polymorphic lymphoid infiltrate) [52]. Further confounding factors are the availability of scarce tissue from core needle biopsies taken at the periphery of the lesions, along with the presence of crush artefact and cellular distortion [69]. Moreover, it has been suggested that DLH/LIP might be the precursor lesion of primary lung MALT lymphoma [70], and the two entities may coexist [5,71], therefore a close follow-up is required upon a diagnosis of LIP due to its risk of eventual progression to lymphoma [52]. Imaging findings disclose a mass in case of both MALT lymphoma and NLH, which is not a typical feature of LIP; the latter presents more often as a diffuse process [52], therefore findings of areas of consolidation, pleural effusion, and large (>11 mm)/growing nodules in a patient with LIP should point toward a lymphomatous transformation [5,8]. Useful morphological clues supporting a reactive lesion, including NLH, are (1) a localized process, lacking signs of architectural disruption (namely, interfollicular lymphoid expansion, follicular ‘colonization’, and lymphoepithelial lesions), (2) a mixed population of CD3+ and CD20+ lymphocytes with expected compartmentalization, (3) absence of expression of CD43 and BCL2 in B cells, (4) a polyclonal pattern at IHC, gene rearrangement, and negativity for t(11;18) [4,52]. Detection of light chain restriction by IHC, flow cytometry, and/or clonality by molecular analysis as well as of MALT1 gene rearrangements by FISH [72] or flow cytometry [73] further support a diagnosis of MALT lymphoma over a reactive inflammatory process. Guinee et al. reported on increased numbers of IgG4 + plasma cells in NLH as compared to MALT lymphoma, thus suggesting that this could be a distinguishing feature between the two entities [74]; obviously, such finding should prompt further examination in order to disclose a form of IgG4-related disease.

At light microscopy, the presence of a heterogeneous population of lymphoid cells rather than a monotonous infiltrate favors the diagnosis of a PP-MZL over a small lymphocyte NHL; since such features may be missed in transbronchial biopsies, IHC is mandatory to establish the diagnosis [68].

SLL typically presents as a monotonous population of small lymphocytes with clumped chromatin, featuring paler areas containing larger lymphocytes with visible nucleoli (i.e., proliferation centers with prolymphocytes and paraimmunoblasts); neoplastic cells coexpress CD20, CD5, and CD23, but lack CD21. Conversely, neoplastic cells in PP-MZL are negative for CD5 and CD23, and residual follicles are CD21+ [4].

MALT lymphoma may mimic FL in cases with abundant reactive lymphoid follicles and extensive follicular colonization. However, a heterogeneous lymphoid proliferation, marginal zone expansion, and lymphoepithelial lesions are rather features of PP-MZL. The immunoprofile of FL is quite typical, showing positivity for germinal center markers (namely, CD10, BCL6, LMO2, HGAL, stathmin), and absence of CD5, CD23, CD43, and cyclin D1. Identification of the rearrangement of the BCL2 gene (t(14;18)(BCL2-IGH)) is particularly helpful in those cases of FL with MALT-like features, namely marginal zone differentiation with monocytoid cells, and in those occasionally lacking CD10 and/or bcl-6 [4,52].

Due to their small size and indented nuclei, the neoplastic cells seen in MCL can overlap the cytological features of centrocyte-like cells in MALT lymphoma. Other morphological characteristics of MCL are a monotonous lymphoid population with vague nodules, hyalinized vessels, scattered naked nuclei from dendritic cells, and histiocytes with eosinophilic cytoplasm (so-called ‘pink histiocytes’). MCL cells coexpress CD20, CD5, and cyclin D1, unlike MALT lymphoma. The differential diagnosis between MALT lymphoma and MCL carries important prognostic implications, due to the more aggressive biological behavior of the latter [4].

Lung is an uncommon location of LPL, which usually involves extra-pulmonary lymph nodes and the bone marrow, unlike pulmonary MALT lymphoma. Additionally, lack of MYD88 mutation further supports a diagnosis of MALT lymphoma with plasmacytic differentiation over LPL.

A MALT lymphoma with extensive plasmacytic differentiation, occasionally featuring intranuclear inclusions (Dutcher bodies) and cytoplasmic crystalline structures (Snapper-Schneider inclusions), may resemble a plasmacytoma; the presence of diffuse sheets of plasmacytoid lymphocytes rather than being located beneath epithelia, their positivity for cyclin D1 and CD56 and negativity for CD20, as well as the absence of monocytoid and centrocyte-like lymphocytes, and lymphoepithelial lesions, all favor the diagnosis of a plasma cell neoplasm over MALT lymphoma [4,52].

In addition, cytogenetic findings by FISH are useful in identifying a FL, or a MCL. Conversely, expression of IRTA1 and MNDA supports a diagnosis of MALT lymphoma [55].

### 2.8. Prognosis and Treatment

Due to its indolent clinical course, disease-specific survival rates at 5 and 10 years for patients with PP-MZL across studies are about 90% and 70%, respectively [29,30,32,37], with a median survival of >10 years [30,37].

Treatment includes surgical resection on localized or peripherally located lesions [40,75], and long watchful waiting, radiotherapy, or single-agent chemotherapy in non-resectable cases, depending on patient comorbidities [40,76,77,78]. Radiotherapy seems to be associated with lower morbidity in comparison with surgery [37,78].

Administration of rituximab may result in long treatment stabilization and symptom reduction [79], yielding a response rate as high as 70%, yet being associated with 36% disease recurrence [79].

A tendency toward an improved progression-free survival (PFS) has been described in patients receiving localized treatments versus systemic therapy, and immunotherapy and immunochemoterapy as compared to chemotherapy alone [32], but it did not achieve statistical significance in terms of overall survival (OS) [76]. The combination of rituximab plus chemotherapy yielded higher response rate and event free-survival according to some authors [32,80]. Systemic therapy is the current treatment of choice for advanced or metastatic MALT-PPL [33,76].

Novel therapeutic strategies are being currently explored in ongoing clinical trials. Noy et al. recently reported on long-term safety and efficacy of single agent ibrutinib, a Bruton’s tyrosine kinase inhibitor, in a cohort of relapsed/refractory MZLs previously treated with rituximab, alone or in combination [81]. Preliminary clinical evidence indicated that PI3K inhibitors copanlisib and umbrasilib showed excellent activity in MZL [82].

Mechanisms of genomic instability, in the form of microsatellite instability, chromosome instability, and aberrant somatic hypermutation, have been described in extranodal MALT lymphomas, sometimes in association with more aggressive clinical outcomes due to the development of a resistant phenotype [83,84,85]; it has been suggested that targeting such specific genetic/epigenetic changes may result in lowering the levels of disease heterogeneity, overcoming such resistant mechanisms, and ultimately increasing effectiveness of targeted treatments [86].

Relapse, as well as concomitant involvement may be seen in approximately 50% of cases within the first years after diagnosis, and in other organs than the lungs as well, such as the stomach, salivary glands, and/or regional lymph nodes [4,8,37,52].

As much as 18% of low-grade MALT lymphoma with concurrent large B-cell lymphoma at diagnosis was reported in one study, and a clonal relation between both neoplasms was disclosed by detection of a concordant light chain restriction [29]. The presence of confluent sheets or large aggregates of large B cells points toward a transformation to DLBCL in up to 20% of cases, resulting in worsening of clinical outcome and change of the treatment protocols. It has been demonstrated that, upon administration of anthracycline-containing chemotherapy regimens, there is substantial overlapping between the survival of patients with PP-MZL alone and of those with DLBCL [29,34,52].

Major prognostic factors associated with survival are age, performance status [30], disease stage, presence of systemic symptoms, coexistence of autoimmune disorders, number and location of pulmonary lesions, presence of paraproteinemia, and type of treatment [20]. An associated pleural effusion or amyloid portends a poorer prognosis [8,40]. An elevated β2-microglobulin level and stage IV classification have been reported as independent predictors of prognosis for MALT lymphoma occurring at all sites [87].

Thieblemont et al. [88] suggested that the addition of three parameters, namely age >70 years, Ann Arbor stage >2, and elevated LDH level, may support risk stratification by discriminating groups of patients with statistically different 5-year PFS rates.

## 3. Primary Pulmonary Diffuse Large B-Cell Lymphoma (PP-DLBCL)

### 3.1. Epidemiology and Clinical Features

PP-DLBCL is the second most common histotype of primary pulmonary lymphoma, accounting for 5% to 20% of all primary pulmonary B-cell lymphomas [8,10], and may occur either de novo, or as large cell transformation from a PP-MZL [29,53]; in the latter case, epidemiological and clinical features overlap with MALT lymphoma [9,28,52]. Male and female patients are equally affected, and median age at presentation is around 60 years (range 30–80 years) [4,28], but it may occur at a younger age in immunosuppressed patients. Although rarely, PP-DLBCL has been described in children as well [89]. In a recent small series of pediatric primary pulmonary lymphomas, three out of four cases were DLBCL, and two of these patients had an underlying immunodeficiency [89].

Patients affected by PP-DLBCL present with ‘B-symptoms’, namely, fever, night sweats, and weight loss of at least 10% of body weight over 6 months [4,7,9]. Further site-specific clinical manifestations are pectoral pain, respiratory distress (in case of bronchial obstruction), dyspnea, cough, and less often hemoptysis [90].

The pathogenesis of PP-DLBCL is poorly understood [28], nevertheless an association with underlying immunosuppression and autoimmune disease has been reported by various authors [28]. It has been suggested that PP-DLBCL may develop in association with long-term methotrexate use, and with cyclosporine A or OKT3 immunosuppression due to solid organ transplantation [9].

Congenital and acquired immune deficiencies may induce the development of DLBCL featuring large EBV+ B cells [27]; EBV+ DLBCL, NOS is enlisted as a distinct disease within the latest WHO Classification, affects mainly elder, immunosuppressed patients, and it has a more aggressive outcome with overall poor prognosis [91]; its pathogenesis is linked to a latent reactivated/chronic persistent viral infection, resulting in the development of a lymphoproliferative disorder [7,91]. The differential diagnosis between EBV+ DLBCL, NOS may be challenging (see below).

### 3.2. Imaging

PP-DLBCL may show a spectrum of radiologic findings, including areas of consolidation, round solid solitary or multiple nodules, a reticular/interstitial infiltrate or ground-glass appearance, with or without mediastinal lymphadenopathy [1,90]. Lesions are mostly bilateral, show sharp borders from the adjacent lung, and tend to be located peripherally in the lower lobe [10], thus transbronchial lung biopsy and ultrasound-guided fine needle aspiration (FNA) are reliable diagnostic methods [92,93,94,95,96]. Typical features at bronchoscopic examination are budding or infiltrative stenosis of bronchi due to tumor invasion [9]. Imaging findings of PP-MZL and PP-DLBCL may sometimes overlap.

Cavitation and/or central necrosis, which presents as low attenuation on multidetector CT scan, is seen in almost half of the cases, and is a much more common feature than in MALT lymphoma [5,10], as well as pleural effusions [9]. Hilar adenopathy may be seen. Imaging findings of consolidation of multiple pulmonary nodules with air bronchograms and a halo of ground-glass shadowing at lesion margins (so-called ‘halo sign’) [92] have been described in pediatric cases as well [89]; such halo sign may be due to peritumoral extension of lymphoma and/or bleeding. FDG-PET usually demonstrates metabolic activity [93].

### 3.3. Pathology

Grossly, PP-DLBCL presents as a tan-white to pale yellow mass, vaguely nodular and with sharp borders [4,52]. Softer areas due to central necrosis, cavitation and areas of hemorrhage are a frequent finding [28] and may sometimes appear as cavities lined by thick walls, mimicking abscesses [20]. The mass may extend and penetrate the visceral pleura or diffusely involve the hilar structures if located centrally.

Microscopically, PP-DLBCL features confluent sheets of medium to large-size dyscohesive lymphoid cells with coarse chromatin and distinct nucleoli, which may show a centroblastic, immunoblastic, or in most cases, combined morphology [3] (Figure 3A,B). Centroblastic-type cells are the most common, displaying oval to roundish nuclei with vesicular chromatin, 2 to 4 juxtanuclear nucleoli, and scant amphophilic to basophilic cytoplasm [3], while immunoblastic features include a prominent central nucleolus with ‘spidery’ chromatin and abundant eosinophilic cytoplasm. PP-DLBCL having >90% immunoblasts are classified as ‘pure’ immunoblastic tumors and are thought to carry a worse prognosis. Some cases may exhibit anaplastic features, with or without Hodgkin/Reed-Sternberg (HRS)-like cells and may pose differential diagnostic issues at light microscopy alone [95]. In all instances, mitoses and apoptotic bodies are frequent, as well as infiltration of bronchial walls, blood vessels, and pleura [52]. Non-neoplastic reactive T-cells may be seen [28]. Local lymph nodes are involved in approximately 50% of cases [4].

Neoplastic cells tend to destroy and replace the entire parenchyma [96]. PP-DLBCL is usually well demarcated from normal lung parenchyma [10], however aggregates of lymphoma cells admixed with fibrin may give the impression of a pneumonic process by filling the alveolar spaces (so-called ‘tumoral pneumonia’) [97]. At core biopsies, the tumor cells may show artefactual spindle morphology and not suggest lymphoma. Areas of residual MALT lymphoma with lymphoepithelial lesions or MALT lymphoma with ‘increased large cells’ may be observed in cases of MALT lymphoma with large cell transformation, lacking a prognostic significance.

### 3.4. Immunohistochemistry and Molecular Findings

PP-DLBCL typically expresses B-cell markers (CD19, CD20, CD79a, PAX5) [3,4] with light chain restriction, while lacking T-cell markers (CD3, CD4); the latter are usually positive in scattered small background T-cells [52] (Figure 3C,D). Even “ghost” tumor cells in areas of extensive necrosis are positive for CD20 or CD79a, which strongly supports a diagnosis of PP-DLBCL [4]. Neoplastic cells may lose their CD20 expression upon treatment with rituximab [28].

In order to subclassify PP-DLBCL as germinal center (GCB) or non-germinal center (non-GCB/ABC) phenotype, a more comprehensive clinical panel including CD10, BCL6, MUM1 should be applied according to Hans algorithm [3] (Figure 3E), as a surrogate of the gene expression profiling, for prognostic and therapeutic purposes. PP-DLBCL shows a non-GCB/ABC phenotype in most cases, like other extranodal DLBCLs, which portends a poorer prognosis [52].

CD30 is more commonly positive in cases with immunoblastic or anaplastic morphology (Figure 3F). EBER in situ hybridization is usually negative.

The Ki-67 proliferation index usually ranges from 40% to 90% [4,53], and it has been suggested that highly proliferative tumors (Ki-67 > 90–95%) should be evaluated for MYC, BCL2, and BCL6 gene rearrangements, and further classified as ‘double-hit’ or ‘triple-hit lymphomas’ accordingly; nevertheless, both forms carry a poor prognosis despite aggressive chemotherapy, with frequent extranodal involvement [52,53]. Those cases which are neither ‘double-hit’ nor ‘triple-hit’ could be however ‘double expressor’ (namely, MYC+ BCL2+).

A monotypic B-cell population is detectable by flow cytometry, nevertheless false negative results due to the presence of abundant necrosis and/or fragility of the large cells should not hamper the diagnosis of PP-DLBCL [28,52].

### 3.5. Differential Diagnosis

Issues in differential diagnosis of PP-DLCBCL are due to its frequent diagnosis on core needle biopsies [52].

PP-DLBCL should be distinguished from a range of high-grade neoplasms, including primary or metastatic poorly differentiated carcinoma, small cell carcinoma, germ cell tumors, melanoma, and sarcoma; the latter diagnosis may be suspected in those cases of DLBCL with artefactual spindle cell morphology [4]. Therefore, a comprehensive immunohistochemical panel of antibodies, including lymphocytic (CD20, PAX5, CD45 (LCA)), melanocytic (HMB-45, Melan-A, MART-1), germ cell (PLAP, OCT3/4, SALL-4), and epithelial (cytokeratins) markers is mandatory.

The differential diagnosis between PP-DLBCL and other pulmonary or mediastinal lymphoproliferative lesions involving the lung, namely high-grade lymphomatoid granulomatosis (LYG), ALCL, primary mediastinal large B-cell lymphoma (PM-LBCL), mediastinal grey zone lymphoma (with features intermediate between B-cell NHL and CHL), and CHL, is more challenging. Both LYG and DLBCL cells express B-cell markers, nevertheless the presence of a large amount of necrosis associated to vasculitis within an angiocentric pattern favors high-grade LYG over PP-DLBCL; however, such features may be missed at biopsy due to undersampling. Moreover, EBER positivity is a common feature of LYG, rather than of PP-DLBCL, along with the common presence of a polymorphic inflammatory background, featuring large amounts of reactive T-cells [3,4]. It has been suggested that a large cell EBER+ lymphoma should be classified as LYG in the proper clinical setting (multiple nodules, no history of transplant and/or use of methotrexate) [52].

The distinction between PP-DLBCL and PM-LBCL is even more difficult, unless a thorough clinical history is provided: young age, female sex, anterior mediastinal mass with secondary lung involvement are typical of PM-LBCL. Both lesions may share the same immuno-morphological features, such as a clear cell morphology and positivity for CD30; however, strong CD30, CD23 and MAL expression favors PM-LBCL over PP-DLBCL [4].

The presence of a polymorphic background with scattered HRS cells should prompt the diagnosis of CHL, whose neoplastic cells have a typical CD79a-/CD20-/CD45(LCA)-/PAX5+ weak/CD15+/-/CD30+/OCT2-/BOB1-phenotype.

ALCL cells are CD30+; however, the lack of T-cell markers and simultaneous presence of B-cell antigens in DLBCL, both usual and anaplastic, allows the distinction between these two lymphomas.

Secondary involvement of the lung by a systemic DLBCL, NOS may occur in advanced stages of disease [52]. Since PP-DLBCL shares the same morphology and immunoprofile with systemic/nodal DLBCL, NOS [98], a correct diagnosis relies entirely on the clinical history and imaging findings [11]. Finally, pulmonary EBV+ DLBCL, NOS should be differentiated from LYG, PEL and DLBCL associated with chronic inflammation (pyothorax) [91].

### 3.6. Prognosis and Treatment

Surgical resection of PP-DLBCL may be performed with curative intent only in localized cases, which are rare occurrences due to its frequent tendency to spread to the mediastinum and extrathoracic organs [4]. Chemo- and/or radiotherapy are usually administered after surgery due to the high local or distant recurrence rate [9,99]. Chemotherapeutic drugs include cyclophosphamide, doxorubicin, adriamycin, vincristine, and prednisone (CHOP) [9,28]. According to a retrospective analysis of over 9000 patients with extranodal DLBCL, CHOP plus rituximab yielded the largest impact on lymphoma-related death in case of spleen, liver, and lung primaries [100].

Median survival times for PP-DLBCL have been reported ranging from 3 to 10 years [4,9]. Conversely, in their large case series Neri et al. found that PP-DLBCL patients treated with conventional CHOP achieved complete response in 77 cases (94%) with 10-year event-free survival as high as 90% [101]. Opportunistic infections in immunosuppressed patients affected by pulmonary lymphomas may further worsen their outcome [7].

EBV positivity, BCL2 translocations, MYC mutations, a “double expressor” phenotype, ABC type according to Hans algorithm, and P53 alterations are all factors portending a poorer prognosis and lower survival rates [3]. First and second phase clinical trials showed that adding bortezomib, lenalidomide and ibrutinib to R-CHOP protocol may improve disease outcome in ABC-type DLBCL [3].

## 4. Lymphomatoid Granulomatosis (LYG)

### 4.1. Epidemiology and Clinical Features

Lymphomatoid granulomatosis (LYG) is an uncommon angiocentric and angiodestructive B-cell clonal lymphoproliferative disorder primarily affecting the lung, with typical clinical, radiological, and histopathological features [52]. It was originally described in 1972 by Liebow et al. [4,102], who reported on 40 patients presenting with nodular lesions of the lungs, reminiscent of lymphoma, as well as of a limited form of Wegener granulomatosis with vasculitis, hence the name. In later years, it has been acknowledged as a lymphoproliferative disease due to the role of EBV its pathogenesis and the detection of oligo/monoclonality, similar to post-transplant lymphoproliferative disorder (PTLD) [19,103].

EBV is essential in the pathobiology of LYG, as correctly suggested in its first report [102], and later confirmed by the detection of EBV-infected B cells in LYG cases by different authors [4,104,105]; it is currently believed that chemokines, released by B lymphocytes upon EBV infection engage T lymphocytes, resulting in vascular injury. The proliferation of EBV-infected B lymphocytes is promoted by host impaired immunologic response, so that a neoplastic cell clone develops [4]. Deficiency of viral proteins prompting cytotoxic T-cell response has been described in LYG [9].

Patients are most often immunodepressed, with underlying acquired or inherited immunodeficiency diseases, such as hypogammaglobulinemia, common variable immunodeficiency, Wiskott-Aldrich syndrome, X-linked severe combined immunodeficiency (SCID), Dedicator of cytokinesis 8 (DOCK8) deficiency, post ablation chemotherapy, solid organ transplantation, HIV infection/AIDS, autoimmune diseases (namely SS, RA, inflammatory bowel disease, and sarcoidosis) [3,9,10,28,97,105], and less often iatrogenic immunodepression (namely, azathioprine, methotrexate or imatinib). Salmons et al. reported on a postomortem case of LYG developed in a patient treated with imatinib for gastrointestinal stromal tumor [106]. In the latter cases, LYG resolution may ensue upon drug discontinuation, further supporting the lymphoproliferative nature of this entity [107,108,109].

LYG is a rare disorder with unknown prevalence [110,111]; however, it occurs more often in Western compared to Eastern countries [3]. Most case have been described as single reports and case series. LYG mostly affects middle-aged male adults (range 30 to 50 years, mean age 46 to 48 years), although it may occur at any age [7,97,105,111].

It is an extranodal disease, mainly affecting the lungs (>90%), therefore a diagnosis of LYG should not be rendered in the absence of lung involvement; the disease may also occur, often synchronously, in the kidneys (45%), skin (25–50%), and central nervous system (25–50%), in decreasing order [3,5,7,36]. Ear, nose, and throat manifestations occur in 10–30% of cases, mimicking granulomatosis with polyangiitis (formerly Wegener granulomatosis (WG)) [112].

Other organs such as liver, spleen, lymph nodes, bone marrow, adrenal glands, heart, eyes, hard palate, gastrointestinal tract, genitourinary tract, are much less often involved [3,52,113], with lymph nodes, spleen and bone marrow being usually unaffected [102]. Disease extension to the central nervous system is associated with a poorer outcome [5,7,97,111].

Almost all LYG patients present with respiratory symptoms, such as fever, persistent productive cough, dyspnea, along with chest discomfort/pain, skin lesions (namely, erythematous plaques, and subcutaneous or dermal nodules of various size primarily located at the trunk and extremities, which may preexist chest radiologic abnormalities) [3,110], potentially life-threatening hemoptysis, malaise, weight loss, arthralgias, myalgias, gastrointestinal symptoms and neurologic alterations; the latter depend on the site of involvement, namely, the central or peripheral nervous system (confusion, paraplegia, hemiparesis, and ataxia), cranial nerves (hearing loss, diplopia, and dysarthria), or neurovegetative system (atonic bladder) [3,4,9,36,110]; the disease may have a remitting and relapsing clinical course over several months to years [102]. Blood count does not show alterations in more than 50% of cases; lymphopenia, leukocytosis, or hypereosinophilia have been seldom described [36,105], as well as polyclonal hypergammaglobulinemia. In case of neural involvement, raised lymphocytes and/or protein concentration may be seen at cerebrospinal fluid analysis [114]. On the other hand, signs of renal insufficiency or proteinuria have not been reported when LYG occurs in the kidney [115]. EBV serology test is usually positive, with EBV viral load often showing minimal and not specifical increase, with a mean of 18 copies/106 genome equivalents [105].

### 4.2. Imaging

Radiologically, typical LYG lesions appear as multiple (range from 5 to 60), bilateral lung nodules of variable size (range, 0.5 cm to >10 cm) scattered throughout mid and lower lobes, with peribronchovascular and interlobular septal distribution [5,116]. Central necrosis/cavitation occurs in up to 30% of cases, often in larger lesions [5,41,97,116]; the latter result from coalescence of single nodules, and may mimic granulomatosis polyangiitis or metastasis [8,9]. Such nodules, which occur in approximately 80% of all cases, may have either ill-defined or sharply demarcated borders, and may contain internal air bronchograms and a peripheral ground-glass halo, as in primary lung malignancy and primary pulmonary lymphoma [116]; they show increased signals on FLAIR and T2-weighted MRI images. Sometimes, they feature a “migratory” behavior (i.e., wax and wane over time) [105]; regressed ones can still be seen as central ground glass opacities with surrounding denser consolidation (so-called “reverse halo sign”), along with new lesions by multidetector CT scan [5]. Pleural effusion is sometimes present [117,118]. Reticulonodular pattern, architectural distortion, and hilar or mediastinal lymphadenopathy are not typical features of LYG [4,8,97,116,117]. Conversely, extrapulmonary organs (e.g., liver, kidneys) usually show focal nodular lesions [52,115].

At MRI, central nervous system involvement is detected as multiple focal lesions in the white matter, deep gray matter or brainstem. Such lesions show punctate linear enhancement, reflecting their perivascular nature. Enhancement of the leptomeninges may be an accompanying feature [119]. FDG-PET images show avid uptake in these lesions and can also demonstrate metabolic activity in the skin, kidneys, and nodes [5,9,10,120], therefore is helpful in evaluating response to therapy [120].

### 4.3. Pathology

Grossly, the lesions have a heterogeneous appearance ranging from tan-white to pale yellow firm nodules to cavitated masses with only a rim of peripheral tan-white viable tissue [3,19]. Centrally located nodules may erode a bronchial wall, leading to mucosal ulceration [4,51]. Although LYG lesions most often present as scattered nodules through the parenchyma, at times they coalesce thus involving an entire lobe [4].

At low magnification, light microscopy findings typical of pulmonary LYG are an angiocentric and angiodestructive pattern along with lymphocytic vasculitis involving small to medium-sized veins and arteries, resulting in varying amount of necrosis according to the grade of the lesion [3,8,119]. Such infarct-like/fibrinoid/coagulative necrosis, typically lacking neutrophils and karyorrhectic debris, is mediated by chemokines induced by EBV [105,121].

A variable number of atypical medium-to-large lymphoid cells, depending on disease grade, is detectable within a polymorphous infiltrate including small lymphocytes admixed with macrophages and scattered plasma cells; such lymphocytes are mostly T-cells, belonging to the T-helper subset [3,47,105,111] (Figure 4A), and may show at time atypical features, in the absence of overt neoplastic morphology [3]. Neoplastic cells are irregular to pleomorphic, and have vesicular chromatin and occasional prominent nucleoli, thus mimicking immunoblasts, plasmablasts, or HRS cells [3,4,28,53,105]. The infiltrate is often sharply demarcated from unremarkable surrounding lung parenchyma [105], yet sometimes peripheral fibroblastic foci, intraalveolar macrophages, and edema could be present, in the absence of a frank organizing pneumonia [105]. Neutrophils, eosinophils and well-formed granulomas are usually absent, the latter however being sometimes detectable at extrapulmonary sites (i.e., skin) [3,4,52]. Regressed or “migratory” nodules are lined by a fibrotic wall surrounded by alveolar edema, hemorrhage, and areas of fibrosis or organizing infarct, containing scant lymphocytic infiltrate and fibrotic blood vessels with recanalized lumen, lacking a muscular wall [4,102]. Dense fibrosis surrounding areas of necrosis are common findings upon treatment [105].

On the basis of the proportion of large atypical EBV+ B cells, necrosis, and the reactive lymphocyte background, LYG can be divided into three grades according to the current WHO Classification of Haematolymphoid neoplasms, with strong prognostic correlation [3,7,97,105,111,119]; they are usually grouped together as low- (grades 1 and 2) and high-grade (grade 3) disease (Table 2), which are associated with better and poorer prognosis, respectively [97]. Such distinction carries important therapeutic implications, since grade 3 disease is treated like diffuse large B-cell lymphoma (DLBCL) [28].

Grade 1 lesions are the least common, featuring a polymorphous lymphoid infiltrate lacking cytological atypia, with absent or rare large B cells, fewer than 5 of them being EBV+ per high-power field (HPF) [3]. The B-cell proliferation is monoclonal in <10% of cells, probably due to the low number of EBV+ atypical cells [4,97,111], and necrosis is absent to minimal. Most patients present at diagnosis with grade 2 lesions; at light microscopy, occasional large atypical lymphoid cells and immunoblasts are detectable within the polymorphous infiltrate, and EBER highlights 5 to 20 of these cells per HPF. The number of EBV+ cells is strikingly variable over the same lesion or multiple lesions [3]. The B-cell proliferation is oligoclonal or monoclonal, more often than in grade 1 (approximately 50%) [4,97,111], and necrosis is common, yet not extensive. In grade 3 lesions, large B cells are abundant, and usually arranged in aggregates and sheets, including more than 50 EBV+ cells in a single HPF. The B-cell proliferation is monoclonal in approximately 70% of cases [4,97,111], and necrosis is usually extensive [10]. Cases of grade 3 LYG without any polymorphous infiltrate are better labeled as EBV+ DLBCL, NOS [3].

Morphological features of increasing grade recapitulate the pathogenesis of LYG [4]. It has been suggested that grade 1 disease may be an early phase where few EBV+ atypical B cells promote cytokine release, resulting in the recruitment of T cells and other inflammatory cells, leading to vascular injury [4]. Later, impaired host immune response favors the proliferation of EBV-infected B cells, driving the progression to grade 2 [4], where the development of an independent malignant cell clone results in grade 3 LYG [4,105]. Interestingly, different neoplastic clones may be observed in different affected organs within the same patient [4]. In support of this step-by-step process hypothesis, the progression rates to LYG grade 3 range from approximately 30% to 70-75% in grade 1 and grade 2 patients, respectively [28].

In order to achieve a correct diagnosis and proper grading of LYG, extensive sampling of the lesions is mandatory. Multiple biopsies should be obtained from each lesion, since disease grade and/or EBV expression may not be homogeneous throughout multiple nodules [36,105]. As a focal disease, LYG may not be adequately sampled via transbronchial or transthoracic lung biopsy [7,9,36], which carry diagnostic rates as low as 30% [52], therefore it has been suggested that surgical lung biopsies, namely lung wedge resections and lobectomies may be performed for small peripheral nodules and larger masses, respectively [7,52,97,119,120].

Cytological examination of BAL may show hypereosinophilia, yet does not provide specific results. In case of involvement of the skin and/or head and neck region (i.e., ear, nose, throat), surgical biopsies may be easily obtained [36,105].

### 4.4. Immunohistochemistry and Molecular Findings

Immunohistochemistry is mandatory for diagnostic confirmation [52]. The large lymphoid cells stain positive for CD45 and have a B-cell phenotype (CD20, PAX5, and CD79a) (Figure 4B). Interestingly, CD20 labels the “ghost cells” in the necrotic areas in grade 3 LYG [105]; these cells, however, are negative by EBER-ISH due to loss of viral RNA under these circumstances [3]. An aberrant CD20/CD43 phenotype has been described [122].

The background small lymphocytes are mostly CD3+ (Figure 4C), with a predominance of CD4+ cells over a lesser amount of CD8+ cells [52].

CD30 expression is variable (30–47% of cases) [105] (Figure 4D), and CD15 is commonly negative [7,97,105,116]. Ki-67 proliferation index is high in grade 3 lesions, paralleling their brisk mitotic count (Figure 4E).

Importantly, the large cells are positive for EBV either detected by ISH (EBER) or by immunohistochemistry (EBV-LMP1) [52] (Figure 4F), consistent with EBV infection in type III latency (EBER+, LMP-1+, EBNA+) [3,105]. However, disagreement exists regarding the most adequate method of identifying and counting the EBV+ B cells [28,105,111]. Katzenstein et al. [111] mentioned the risk of misdiagnosing LYG grade by using an EBER ISH-based method, due to the high interlaboratoratory variability of its sensitivity rates. Accordingly, cases of EBER- LYG have been reported; moreover, small lymphocytes can be labeled by EBER, resulting in an overrated count. To overcome this issue, a more reliable and reproducible grading based on the number of CD20-positive atypical cells has been suggested. Interestingly, LYG lesions may be negative for EBER in up to 50% of skin lesions [105,110].

The current grading system is also hampered by the presence of nodules of different grades within the same biopsy specimen, or between different specimens from the same patients, due to the intrinsic heterogeneity of LYG lesions [105].

Light and heavy chain rearrangements by clonality analysis may be useful, yet it is not necessary for the diagnosis [8]; monotypical expression of immunoglobulin light chains may be detectable in plasmacytoid cells within the infiltrate [3].

### 4.5. Differential Diagnosis

From a clinical standpoint, the presence of multiple lung nodules in an immunosuppressed patient provides an initial hint for diagnosing an infectious process; the lack of response to antibiotic and antifungal treatment leads to further diagnostic evaluation through examining a tissue biopsy of the lesion [52].

The differential diagnosis of LYG includes a broad spectrum of entities according to the grade of the lesion and amount of necrosis [4,36,52]. In this setting, important hints are provided by radiological correlation, in that lymphoproliferative disorders usually present as solitary lung masses, and at advanced stage of disease; the occurrence of lymph node involvement alone prevents a diagnosis of LYG, which is mostly an extranodal disease [123].

Detection of vasculitis, an angiocentric pattern, and necrosis are mandatory, since these features are not typical of lymphomas. In case of extensive necrosis, a pulmonary infarct, a vasculitic process (such as, WG, necrotizing pneumonia (aspergillosis or due to Pseudomonas, Klebsiella, etc.), sarcoidosis, necrotizing sarcoid granulomatosis, or a necrotic primary or secondary tumor to the lung should be taken into account [8,52]. Immunohistochemistry is ultimately required to differentiate among these lesions.

LYG lesions with grade 1 or 2 features enter the differential diagnosis of polymorphic lymphomas/lymphoproliferative disorders (CHL, T-cell/histiocyte-rich DLBCL (TCHR-DLBCL), polymorphic PTLD, and peripheral T-cell lymphoma (PTCL, NOS)) and WG [4,52]; the latter is easily distinguishable from LYG by the presence of typical inflammatory granulomas [28].

The expression of CD30 and EBER in large cells with atypical nuclei may lead to a misdiagnosis of CHL; additionally, HRS cells may sometimes be CD20-positive. However, EBER+ cells in LYG are of variable size, ranging from small to large, unlike CHL [3,105,119]; the latter features CD15+/CD3-/CD20-/CD45- cells [28]. Moreover, neutrophils, eosinophils and well-formed granulomas are not typically seen in the tumor anlage.

The large B-cells in TCHR-DLBCL are CD20+, but lack EBER expression, although a related subtype of EBV+ DLBCL exists which has been defined TCHR-like due to inherent similarities in morphology; moreover, TCHR-DLBCL contains a huge amount of CD68 + macrophages [4].

On the other hand, DLBCL, ALCL, NK/T-cell lymphoma, nasal type (NKTCL-NT), and monomorphic PTLD can mimic grade 3 LYG [4,52].

EBER positivity along with the presence of a polymorphous background identify an otherwise straightforward PP-DLBCL as LYG grade 3, in view of a proper clinical history, although occasional scattered EBER+ cells may be detected in DLBCL. When present, necrosis in DLBCL is neither as extensive as in LYG nor angiocentric [105].

In the posttransplant and/or immunomodulatory setting a lesion with morphological features similar as grade 3 LYG should be better classified as PTLD or an iatrogenic immunodeficiency-associated lymphoproliferative disorder, respectively [111,124].

The presence of overlapping features with NKTCL-NT, such as vasculitis, EBV association and angiocentric growth pattern with variable amount of necrosis [3,4], may challenge the diagnosis of LYG, since NKTCL-NT can occur in lungs as well. Useful clues in this setting are (1) the absence of apoptotic/karyorrhectic debris (also called “dirty-type” necrosis) [3,105], (2) the higher prevalence in Western countries than in Asia, and (3) the presence of large atypical cells of B-cell origin lacking positivity for T-cell and/or cytotoxic markers (such as, CD3, CD8, CD56, TIA-1, granzyme, and perforin) and T-cell receptor gene rearrangement, all of which are features of LYG rather than NKTCL-NT [3,52,105,119]. Unlike other T-cell lymphomas, namely PTCL, NOS, and ALCL, LYG may show a T-cell phenotype only in small cells and is positive for EBER [28,52].

### 4.6. Prognosis and Treatment

LYG has a poor prognosis overall, with median survival as high as 2–4 years [97,111,116,117,119], although it may vary greatly. In a seminal retrospective study on 152 patients, Katzenstein et al. reported a mortality rate as high as 63.5%, mostly within the first year after the diagnosis of LYG [125]. A progressively decreasing survival is associated with pulmonary involvement, systemic involvement, and CNS involvement, respectively. Further clinical factors associated with poor prognosis are age <25 years, hepatosplenomegaly, leukopenia, persistent fever [36,105,121,125]. Death ensues due to respiratory insufficiency, hemoptysis, neurological complications, and infection, in decreasing order [105,121,125]. Post-mortem studies on LYG patients revealed the frequent involvement of multiple organs, including skin, CNS, lymph nodes, spleen, liver, kidneys, adrenal glands, heart, and rarely testes and eyes [20].

Treatment is administered according to the patient’s immunosuppression status and severity of symptoms, along with the disease stage and histological grade [3,105,111,119]; the latter is associated with the risk of transformation to EBV+ DLBCL and more aggressive diseases [8,28].

In case of iatrogenic impairment of the immune system, discontinuation of previous therapy is mandatory. In most low-grade lesions, careful follow-up is warranted since spontaneous remission has been reported, especially in grade 1 LYG [5,119,125], and surgery or radiotherapy alone are optimal therapeutic options for localized LYG [126]. Conversely, more aggressive therapy is considered for patients with high-grade disease [123,127].

Treatment options for grade 3 disease include high dose steroids, rituximab, combined chemotherapy regimens (namely, CHOP, ICE (Ifosfamide, Carboplatin, Etoposide), and hyperCVAD (Cyclophosphamide, vincristine, doxorubicin and dexamethasone, along with methotrexate and cytarabine)), immunomodulators, usually along with other medications in order to evade immune system impairment [4,8,36,105,111,119,121]. Of them, treatment based on rituximab alone has given inconsistent results [119,120,128], while the use of combined regimens yielded a significant improvement in survival (70% 5-year survival rate) [3]. Usually, after an initial symptomatic improvement, recurrence may ensue over time.

Upon the use of interferon-alpha 2b in low-grade disease, better PFS and higher complete remission rate of 60% ensued according to a National Cancer Institute (NCI) study; such complete remission in most patients with CNS involvement resulted in discarded intrathecal chemotherapy or whole brain radiation [103]. Administration of EPOCH-R (rituximab, prednisone, etoposide, vincristine, cyclophosphamide, and Adriamycin) regimen to high-grade LYG patients prompted increased overall survival of 68% with a median of 4 years [3,119]. A further second or third-line therapeutic option may be autologous or allogeneic stem cell transplantation [129].

## 5. Primary Effusion Lymphoma (PEL)

### 5.1. Epidemiology and Clinical Features

A subset of primary pleural large B-cell lymphoma featuring human herpes virus-8 (HHV-8) positivity, and EBV coinfection in half of the cases [130], has been defined as primary effusion lymphoma (PEL) according to the latest WHO classification [3,11].

PEL is a rare lymphoma which accounts for 0.5% of all lymphomas and 1–8% of HIV-associated NHLs [131,132]; most patients are HIV+, usually with AIDS and Kaposi sarcoma, or any underlying immunodeficiency, in HHV8-endemic areas (see below) [133,134]. Although extremely rare, cases lacking HIV and HHV-8 have been documented [135].

PEL is typically located in mesothelial lined body cavities (pleura, pericardium, or peritoneum), and it should be distinguished from solid lymphoma with secondary effusion localization [3,136]. However, PEL has also been reported in a pure or mixed solid form (the so-called “extracavitary PEL”) [137,138], and either at first diagnosis or as a relapse of classic PEL [131]. PEL commonly arises in young to middle aged immunosuppressed males [136].

Recurrent pleural effusion is present in 85% of cases and is associated with ascites in 50% of cases [139], showing a wide range of signs and symptoms, depending on the site of occurrence and amount of the effusion. Patients mostly present with fever, altered performance status, and dyspnea [140]. A secondary cardiac tamponade may develop in case of pericardial involvement [140]. Further clinical manifestations may occur, namely hepatosplenomegaly (in almost 65% of cases) [140], concomitant Kaposi sarcoma (25–100% of cases) and multicenter Castleman disease (MCD, 9–50% of cases); peripheral lymphadenopathy is rare. Other organ-specific symptoms, such as neurological deficits, should raise suspicion of an extracavitary disease [139]. Peripheral blood analysis mostly shows anemia, along with thrombocytopenia and hypoalbuminemia in almost half of the cases [139], raised LDH levels [135], and mean CD4 count up to 150–200/mm^3^ [140].

### 5.2. Imaging

CT scans usually reveal unilateral effusions, sometimes with a small pleural thickening, in the absence of detectable tumor lesions or mediastinal adenopathy [36]. Concomitant pericardial or peritoneal effusion may occur [141].

### 5.3. Pathology, Immunohistochemistry and Molecular Findings

Ultimately, the diagnosis is based on the cytological examination of the serohematic pleural fluid, rather than on tissue biopsy evaluation [136,140]; a cell block should be obtained in order to perform ancillary techniques (IHC and ISH) [134]. PEL cells are usually not detected at bone marrow trephine biopsy, which may show haemophagocytosis [139,140].

At light microscopy, neoplastic cells are arranged in sheets; they are large, sometimes immunoblastic, showing abundant hyperbasophilic cytoplasms, polylobular nuclei with irregular contours, and prominent nucleoli (Figure 5A). Pleomorphism may be striking, and RS-like cells may be identified in some cases [3]. The mitotic index is high.

The classical immunophenotype of PEL is CD45(LCA)+ (Figure 5B) but CD19-/CD20-/CD22-/CD79a-/PAX5-. Neoplastic cells may express aberrant T-cell markers, such as CD2, CD3, CD4, and/or CD5 [3,142]. They may coexpress the plasma cell markers CD138, CD38, vs38, MUM-1, without expression of surface immunoglobulin [143,144], and the activation markers CD30 and HLA-DR [144]. Clonality analysis supports its B cell lineage despite the lack of B-cell antigen expression, which are at least partially expressed in the extracavitary form of PEL [137,144].

HHV8 infection may be disclosed by IHC with latent nuclear antigen antibody (LANA1), or by PCR evaluation of the viral load in pleural fluid. In blood, the HHV8 viral load is generally >4 log copies/105 peripheral blood mononuclear cells, and there is a 3:4 log ratio between pleura and blood [145]. Rare cases of HHV8- PEL may occur, some of them showing an immunophenotype similar to HHV8+ PEL (i.e., lack of B-cell markers) [146].

When present, EBV coinfection is confirmed by EBER staining or PCR analysis of the pleural fluid rather than IHC, since staining with latent membrane protein is negative [139]. Interestingly, HHV-8-PEL [146] are usually EBV-.

Cytogenetic analysis shows a complex karyotype without common or recurrent chromosomal abnormality [147]. Specifically, PEL is not associated with alterations of the c-MYC gene [148].

### 5.4. Differential Diagnosis

Pleural effusion may occur in immunocompromised patients due to Kaposi sarcoma, multicenter Castleman disease, PEL, or in the setting of an infectious disease such as tuberculosis; in such cases, cytological examination of pleural fluid is useful in discriminating among these entities, along with the correct clinical context [36].

The expression of HHV8 in neoplastic cells is a helpful clue in the differential diagnosis with other lymphomas [134]; however, extracavitary PEL should be differentiated from other EBV+ or HHV8+ B-cell lymphomas. Plasmablastic lymphoma (PBL) shares similarities with PEL, namely the lack of B-cell markers and the positivity for plasma cell markers; however, c-MYC anomalies are not found in PEL, and PBL cells are HHV8- [149].

In cases of PEL with more anaplastic morphology, the differential diagnosis is with CHL and ALCL. Expression of CD45 and absence of CD15 in neoplastic cells support a diagnosis of PEL versus CHL [137]. ALCL may be suspected in those cases expressing multiple T-cell markers such as CD3, CD4, CD5; however, the B-cell origin of the neoplastic cells is revealed using other ancillary techniques, namely ISH for kappa and lambda light chains and PCR-based studies for IgH and TCR rearrangements [134].

### 5.5. Prognosis and Treatment

Its aggressive clinical behavior and overall resistance to conventional B-cell lymphoma therapeutic drugs (such as, rituximab) result in a poor outcome in most cases [147], with a median survival interval <1 year, and a 39% 1-year survival [139]. Remission may be obtained upon optimization of antiretroviral therapy, which has shown some antitumoral activity [138], and/or using CHOP with or without methotrexate [139]; however, this latter treatment is associated to significant morbidity. Some authors reported on the use of other drugs, including bortezomib, rapamycin, anti-vascular endothelial growth factor therapy, or interferon in single cases [139,150,151]. Palliative treatments encompass pleurodesis with talc or intrapleural cidofovir [152,153]. Lack of antiretroviral therapy at diagnosis, the number and type of involved cavities [149], LDH level and CD4 positivity [154] have been reported as putative prognostic factors, yet with conflicting results [131]. CD20 expression has been identified as a favorable prognostic factor, along with age, in the rare subset of HHV8-PELs [146].

## 6. Intravascular Large B Cell Lymphoma (IVLBCL)

### 6.1. Epidemiology and Clinical Features

Intravascular large B cell lymphoma (IVLBCL) is a rare subtype of extranodal lymphoproliferative disorder featuring proliferation of neoplastic cells within the lumina of capillaries and small to medium-size vessels [3]. IVLBCL has a poor clinical outcome due to the fast and wide dissemination of tumor cells to several organs, however lymph nodes are usually spared [3].

IVLBCL affects men and women equally, with a median age of 67 years. Its clinical presentation, organ involvement and prognosis may vary depending on the patient’s country of origin, and two clinical variants have been recognized: (1) a classic form, mostly occurring in Western countries, whose symptoms depend on the affected organs (most commonly CNS and skin), and (2) a haemophagocytic syndrome-associated form, more typical of Eastern countries, with multisystemic involvement, hepatosplenomegaly and pancytopenia [155,156]. However, primary pulmonary symptoms are not common [76], as well as primary presentation in the lungs [9], although pulmonary IVLBCL is often found at postmortem examination [156].

A further clinical pattern affects only the skin, is usually diagnosed in western women, and carries a better prognosis [3]. The pathogenesis of IVLBCL is unknown [157]. Several hypotheses have been made in order to unveil the mechanisms used by tumor cells to remain within the vessel lumina [158]; for instance, lower levels of molecules involved in trans-vascular migration, namely CD29 (integrin 1ß) and CD54 (ICAM1) has been detected in IVLBCL cells [28]. Likewise, neoplastic cells lack the expression of adhesion molecules such as CD18 [156]. Finally, the interaction between CXCR3 expressed by IVLBCL cells and its ligand CXCL9 on endothelial cells might underlie a further pathogenetic mechanism [157].

Molecular analyses disclose monotypical rearrangements of immunoglobulin genes [28]. No specific cytogenetic alterations have been reported, nonetheless structural abnormalities at chromosomes 1 (1p), 6 and 18 (trisomy 18) have been identified [28].

### 6.2. Imaging

At imaging, the most typical presentation is as bilateral ground-glass, centrilobular nodules, or interstitial opacities, rather than a discrete mass [9,76,157], or no abnormalities at all. FDG-PET scan may be more informative by demonstrating diffusely increased pulmonary tracer uptake in the absence of corresponding abnormality on CT images [158].

### 6.3. Pathology

Grossly, IVLBCL diffusely invades extranodal sites of origin, including lung [28]. Involved tissues may appear hemorrhagic, necrotic, or show vascular thrombosis, although sometimes macroscopic changes are undetectable [20].

At light microscopy, there is diffuse intravascular proliferation of atypical B cells [20], which fill and distend arterioles, veins and capillaries [20] (Figure 6A), and occasionally penetrate into the vessel wall. Thrombotic and ischemic complications may ensue in the affected organ [9]. At low magnification, cases showing striking diffuse expansion of alveolar capillaries may appear as interstitial widening, mimicking non-neoplastic diseases such as interstitial pneumonia or interstitial lung disease [4]. Such neoplastic cells are large, with vesicular nuclei, prominent nucleoli, and brisk mitotic activity [3]. Rarely, IVLBCL cells show marked anaplasia or small size [3]. Extravascular involvement by lymphoma cells may be seen, especially in the perivascular areas [50].

Four growth patterns have been identified: (1) a non-cohesive pattern (tumor cells are freely floating with the vascular lumen); (2) a cohesive pattern (tumor cells are arranged in aggregates which fill and occlude the vascular lumen); (3) a marginated pattern (tumor cells adhere to endothelia without forming neoplastic thrombi); and (4) a tumor-associated pattern (tumor cells invade vascular structures within another tumor, partially sparing vessels in the normal surrounding parenchyma.

Hemorrhagic area, necrosis and fibrin thrombi may be occasionally seen [3,20]. Haemophagocitosis is a typical finding in the haemophagocytic syndrome-associated form of IVLBCL [28].

### 6.4. Immunohistochemistry and molecular findings

Tumor cells express CD45 along with B-cell markers, including CD20 [20] (Figure 6B,C), with coexpression of BCL2 and BCL6 in 90% and 25% of cases, respectively [47]. CD5 and CD10 are positive in 38% and 13% of cases, respectively [3], and coexpression of both markers may be seen. Expression of CD5 in tumor cells is more frequent in LBCLs devoid of an intravascular component and does not portend an association with chronic lymphocytic leukemia or mantle cell lymphoma [20]. Moreover, most CD10-IVLBCLs are IRF4/MUM1+ [3]. Most cases are of non-GBC/ABC origin and EBV- [52]. There is high Ki-67 proliferation index [47] (Figure 6D).

CD34 immunostaining for endothelial cells may be useful in challenging cases, although this marker is not present in routine panels of antibodies [3]. Although very rare, some forms of intravascular lymphomas of T-cell origin have been reported [159].

### 6.5. Differential Diagnosis

The diagnosis of IVLBCL is often challenging, due to its not-specific clinical presentation and peculiar morphology [20]. Several entities, including LYG, angiocentric lymphoma, sarcoma, secondary lymphoproliferative lesions, and lymphangitic spread to the lung by acute leukemia, carcinoma, or melanoma should be taken into account in the differential diagnosis [52,157], as well as non-neoplastic lesions such as interstitial diseases [28]. IVLBCL cells share cytological, and sometimes immunophenotypical features with DLBCL, nonetheless the latter is easily distinguishable due to its capacity to form a distinct tumor mass [47].

### 6.6. Prognosis and Treatment

IVLBCL has an overall poor prognosis, except for the cutaneous forms, due to its intrinsic aggressive behavior and to its non-specific presentation and/or imaging findings resulting in a delayed diagnosis [3]. Nonetheless, early administration of anthracyclin or rituximab yields a better outcome compared to patients not receiving this therapy [20], with achievement of a complete response in about 50% of cases [9]. Even upon treatment with rituximab, patients may experience CNS relapse and neurolymphomatosis in 25% of cases; no prognostic factors have been identified to date [3].

## 7. Conclusions

Primary pulmonary B-cell lymphomas represent the vast majority of lymphomas arising primarily in the lung. They represent a heterogeneous group of diseases with respect to their clinical, radiological, pathological, and molecular features, resulting in different treatment protocols and outcomes. Knowledge of their clinical and pathological features is of paramount importance, since early diagnosis relies on the alertness of clinicians, mainly radiologists, pneumologists and thoracic surgeons, resulting in a proper diagnostic workup. Recently, novel treatment options based have been described, namely tyrosine kinase inhibitors, PI3K inhibitors, and molecular-targeted therapies, which showed promising results at preliminary clinical evidence. The management of PP-BCLs requires a multidisciplinary approach including radiologists, pneumologists, thoracic surgeons, pathologists, hemato-oncologists, and radiation oncologists, in order to achieve a correct diagnosis and risk assessment.

## Figures and Tables

**Figure 1 cancers-13-00415-f001:**
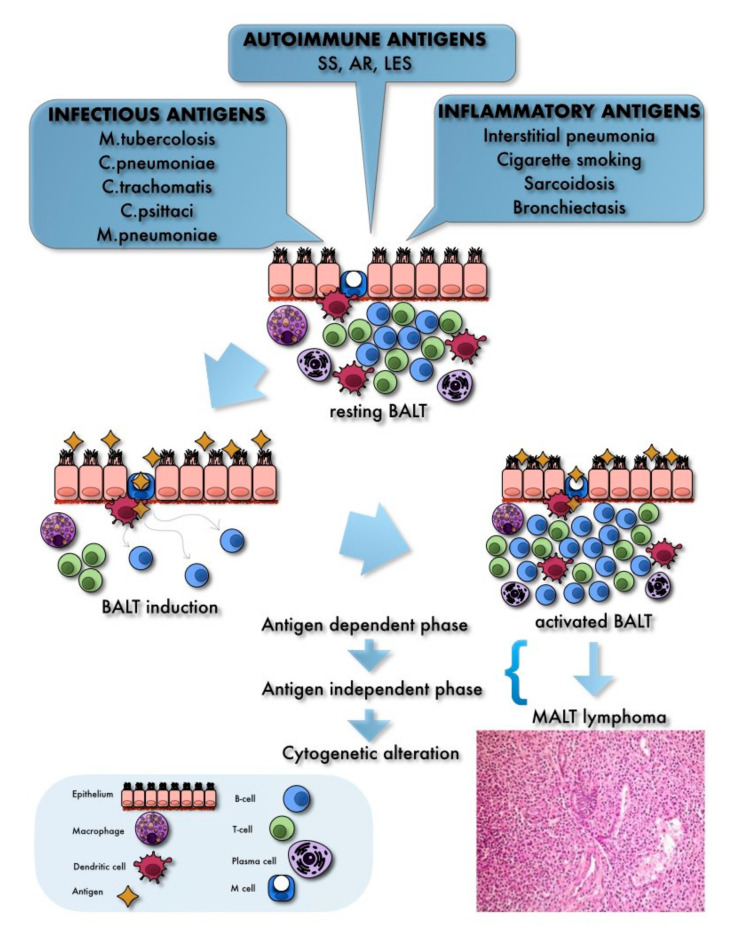
The stepwise progression from BALT to MALT lymphoma.

**Figure 2 cancers-13-00415-f002:**
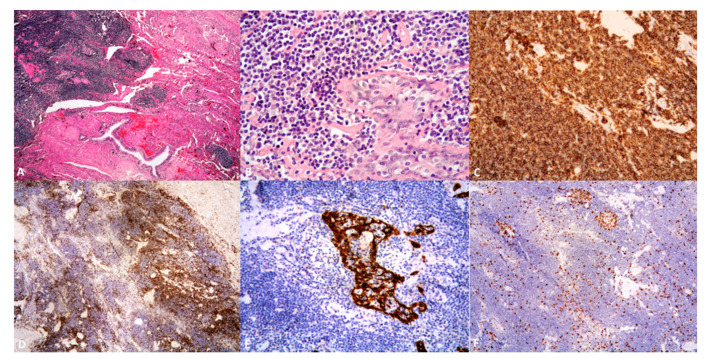
Primary pulmonary MALT lymphoma. (**A**,**B**). A dense infiltrate of mostly centrocyte-like lymphocytes effaces the lung parenchyma (A H&E stain, 40×; B H&E stain, 400×). (**C**,**D**). Neoplastic cells are CD20+ with a minor population of reactive CD3+ T-cells; both antibodies feature a membranous staining ((**C**) CD20 immunostain, 200×; (**D**) CD3 immunostain, 100×). (**E**). Lymphoepitelial lesions due to neoplastic cells infiltrating the epithelium are highlighted by cytokeratins (CK-AE1/AE3 immunostain, 200×). (**F**). The proliferative index, i.e., the percentage of Ki-67 stained nuclei, is low in the neoplastic cells, but higher in residual germinal centers (Ki-67 immunostain, 100×).

**Figure 3 cancers-13-00415-f003:**
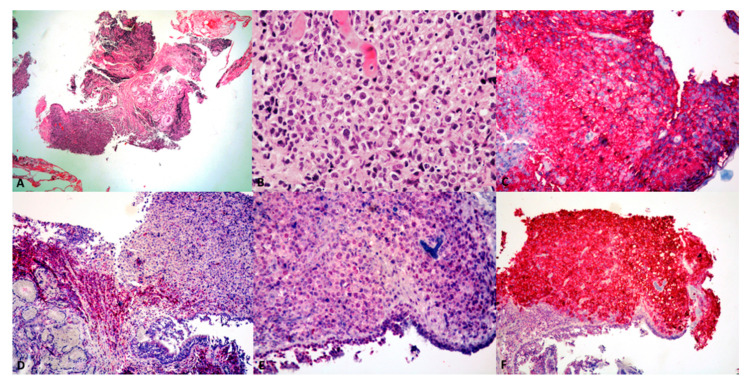
Primary pulmonary diffuse large B-cell lymphoma. (**A**,**B**). At low and high power, respectively, a heavy infiltrate of large tumor cell, the majority with centroblastic morphology (A H&E stain, 40×; B H&E stain, 400×). (**C**,**D**). Neoplastic cells show diffuse and intense expression of CD20, with fewer reactive CD3+ T-cells ((**C**) CD20 immunostain, 200×; (**D**) CD3 immunostain, 100×). (**E**). In this GCB-type PP-DLBCL there is diffuse nuclear expression of BCL6 (BCL6 immunostain, 200×). (**F**). Tumor cells are intensely positive for CD30 in another case with immunoblastic morphology (CD30 immunostain, 100×).

**Figure 4 cancers-13-00415-f004:**
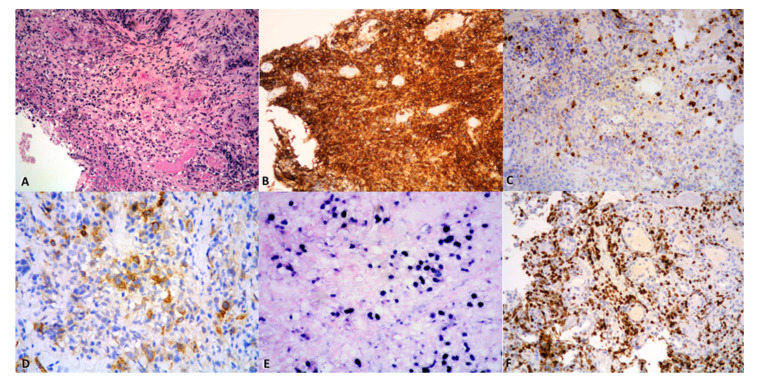
Lymphomatoid granulomatosis. (**A**). A number of atypical medium- to large- lymphoid cells within a polymorphous infiltrate (H&E stain, 200×). (**B**,**C**). Neoplastic cells show diffuse and intense expression of CD20 in this case of grade 3 LYG, with fewer reactive CD3+ T-cells ((**B**) CD20 immunostain, 200×; (**C**) CD3 immunostain, 200×). (**D**,**E**). Few cells show positivity for CD30 in another case of grade 2 LYG (CD30 immunostain, 400×); the same cells are intensely positive for EBER (EBER ISH, 400×). (**F**). Grade 3 LYG has a high proliferation index, i.e., >50% tumor cells showing nuclear immunoreactivity (Ki-67 immunostain, 400×).

**Figure 5 cancers-13-00415-f005:**
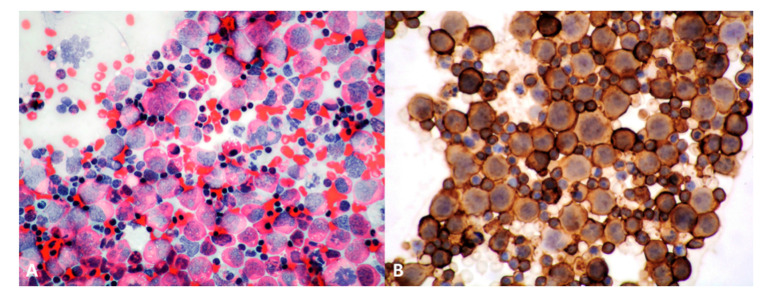
Primary effusion lymphoma. (**A**). Large tumor cells are abundant in this pleural fluid sample (Papanicolau stain, 400×). (**B**). There is diffuse and intensemembranous and cytoplasmic staining for CD45 (Leukocyte Common Antigen) (CD45 immunostain, 400×).

**Figure 6 cancers-13-00415-f006:**
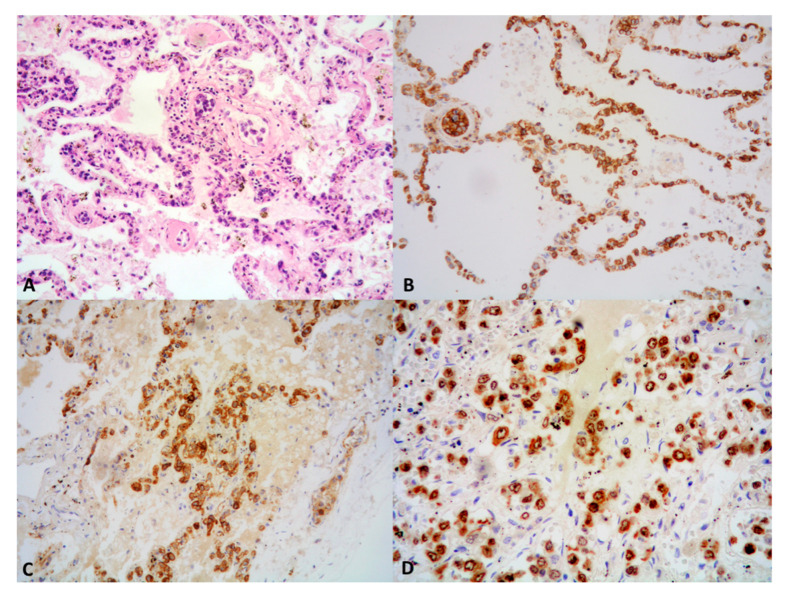
Intravascular large B-cell lymphoma. (**A**). Large neoplastic cells fill the vessels within the interstitial spaces (H&E stain, 200×). (**B**–**D**). Tumor cells are positive for CD45 (**B**), CD20 (**C**), and show a high proliferative index, i.e., >90% tumor cells showing nuclear immunoreactivity (**D**) ((**B**) CD45 immunostain, 200×; (**C**) CD20 immunostain, 200×; (**D**) Ki-67 immunostain, 400×).

**Table 1 cancers-13-00415-t001:** Summary of the main features of primary pulmonary B-cell lymphoproliferative disorders.

Histotype	Main Imaging Findings	Pathology	Immunophenotype	Behavior	Treatment
PP-MZL	Single/multiple nodules/masses with bronchovascular distribution (‘butterfly sign’, ‘bronchogram and CT-angiogram sign’)	Reactive lymphoid follicles + diffuse centrocyte-like lymphocytes + lymphoepithelial lesions; (often monotypic) plasma cells	CD19+, CD20+, CD22+, CD79a+, IRTA-1+, MNDA+, CD3-, CD5-, CD10-, BCL6-, cyclin D1-	Mostly indolent	Surgery + watchful waiting, radiotherapy, chemotherapy (+/− rituximab)
PP-DLBCL	Areas of consolidation, single/multiple nodules, reticular/interstitial infiltrate, ground-glass appearance; mostly bilateral	Sheets of medium to large-sized lymphoid cells with centroblastic, immunoblastic, or mixed morphology	CD19+, CD20+, CD79a+, PAX5+, CD3-, CD4-	Aggressive	Surgery + chemo/radiotherapy (R-CHOP)
LYG	Nodules of variable size, masses, areas of consolidation with bronchovascular and interlobular septal distribution	Polymorphous lymphoid infiltrate +/− large atypical cells, angiocentric and angiodestructive pattern, necrosis	CD20+, CD79a+, PAX5+, EBV(LMP-1)+, EBER+	Variable, may be aggressive	According to disease stage and grade, and to patient’s immunosuppression status
PEL	Unilateral effusions +/− small pleural thickening	Sheets of large atypical cells	CD45+, HHV8+, CD19-, CD20-, CD22-, CD79a-, PAX5-, CD2+/−, CD3+/−, CD138+/−, MUM-1+/−, EBV(LMP-1)-, EBER+ (usually negative in elderly HIV negative)	Aggressive	Antiretroviral therapy +/− CHOP +/− MTX
IVLBCL	Bilateral ground-glass, centrilobular nodules/interstitial opacities	Diffuse intravascular proliferation of atypical B cells	CD45+, CD20+, BCL2+	Aggressive	Anthracyclin, rituximab

**Table 2 cancers-13-00415-t002:** Grading of lymphomatoid granulomatosis.

Grade	Background Infiltrate	Large Atypical Cells	Necrosis	EBER+ Cells/HPF
Grade 1	Polymorphous lymphoid infiltrate	Absent to rare	Absent to focal	<5
Grade 2	Polymorphous infiltrate with few large cells or immunoblasts	Scattered or arranged in small clusters	Common	5–20, occasionally >50
Grade 3	Focal polymorphous infiltrate with large atypical cells	Arranged in clusters or large aggregates; evidence of few Hodgkin cells	Extensive	>50

## Data Availability

Data sharing not applicable.

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
