# Peer review of "Primary Pulmonary B-Cell Lymphoma: A Review and Update"

_cancers, 2021, doi:10.3390/cancers13030415_

Round 1

Reviewer 1 Report

In this review, Francesca Sanguedolce et al summarized various aspects of Primary Pulmonary B-cell Lymphoma. The manuscript covers several aspects of the subject and is well organized. The paper is an update on the topic.

However, some paragraphs (2.5, 2.6, 2.7…) are too long and need to be rewritten.  

For the clarity of the paper, the authors could summarize all the data in a table.

The role of genomic instability and clinical outcomes should be more described regarding the new therapeutic strategy.

The conclusion is too short and the authors should describe the new therapeutic strategy and the possible of the diagnosis of these pathologies in early stage.

Authors need to pay attention to the presentation of figures, particularly details in regards to clarity of immunohistochemistry.

Reviewer 2 Report

The authors have prepared a comprehensive review of the subject, which is well-organized, thoroughly referenced, and expertly written. I have only minor suggestions:

1) The sections on treatment of the various types pulmonary lymphomas are generally limited to conventional, standard-of-care chemotherapy. There should be some mention of the options currently available with clinical trials, and the prospects for improved efficacy/toxicity ratio potentially available with immunotherapy and targeted therapy. E.g., it was recently reported that single-agent BTK inhibition is effective in extranodal marginal zone lymphomas, even relapsed (PMID: 33227125).

2) With so much information presented in text form, it might be useful to have one or more tables that present the key features of the various types pulmonary lymphomas.

3) In line 563, "alike other extranodal DLBCLs" should be "like other extranodal DLBCLs".

Reviewer 3 Report

General evaluation:

Sanguedolce et al. provide thorough review of B-cell lymphomas that primary develop in the lung. These include pulmonary MALT lymphoma, diffuse large B-cell lymphoma, and lymphomatoid granulomatosis (LYG), in addition to primary effusion lymphoma and intravascular large B-cell lymphoma (IVLBCL). Although earlier papers on similar topics are found in the literature, the current article is certainly of value to the readers in this field. Nevertheless, this reviewer suggests several points to be modified before publication, as listed below.

Comments:

Page 3, line 109: Clonal expansion of B-cells resulting from chronic exposure to specific antigenic stimuli may be supported by biased usage of certain immunoglobulin heavy chain gene families (Xia H et al. Hum Pathol, 2011;42:1297).

Page 8, line 357: The authors might refer briefly to the mutational landscape of MALT lymphoma revealed by next generation sequencing technology (Cascione L et al. Haematologica, 2019;104:e558).

Page 10, line 458: For relapsed/refractory MALT lymphoma, a B-cell receptor signaling inhibitor, ibrutinib, and lenalidomide in combination with rituximab are currently approved. PI3K inhibitors may be listed in the treatment option in the near future.

Page 11, line 476: The authors misreferred to the Thieblemont’s paper (#83 instead of #82).

Page 15, line 668: The authors might refer to the paper by Salmons N et al. (J Clin Pathology, 2007;60:199-201), reporting on a patient with GIST who developed LYG after imatinib treatment.

Page 23, line 1050: FDG-PET imaging is capable of predicting IVLBCL by increased tracer uptake within the lung (Spencer J et al. Radiol Case Rep, 2019;14:260).

Page 23, line 1076 and Figure 6: CD45 immunohistochemistry (IHC) is not necessarily of value to the diagnosis of IVLBCL, as lymphoma cells of this disease invariably express CD20. Instead, CD34 IHC is capable of confirming intravascular growth of lymphoma cells.

References: #52 and #145 are missing.

Round 2

Reviewer 1 Report

thank  you for the sevaral changes in the manuscript